# ReMixer: Object-aware Mixing Layer for Vision Transformers and Mixers

**Hyunwoo Kang**[*]**, Sangwoo Mo**[*]**, Jinwoo Shin**
Korea Advanced Institute of Science and Technology (KAIST)
`{hyunwookang,swmo,jinwoos}@kaist.ac.kr`

## Abstract

Patch-based models, e.g., Vision Transformers (ViTs) and Mixers, have shown impressive results on various visual recognition tasks, exceeding classic convolutional networks. While the initial patch-based models treated all patches equally, recent studies reveal that incorporating inductive biases like spatiality benefits the learned representations. However, most prior works solely focused on the position of patches, overlooking the scene structure of images. This paper aims to further guide the interaction of patches using the object information. Specifically, we propose ReMixer, which reweights the patch mixing layers based on the patch-wise object labels extracted from pretrained saliency or classification models. We apply ReMixer on various patch-based models using different patch mixing layers: ViT, MLP-Mixer, and ConvMixer, where our method consistently improves the classification accuracy and background robustness of baseline models.

## 1 Introduction

Patch-based models, i.e., models that process an input image as a sequence of visual patches, have arisen as a new paradigm of neural networks for visual data, alternating the prior standard convolutional neural networks (CNNs; LeCun et al. (1998)). Remarkably, patch-based models have achieved state-of-the-art results on various computer vision tasks by favoring the scaling properties (Dosovitskiy et al., 2021; Zhai et al., 2021). They also merit various advantages, including out-of-distribution generalization (Naseer et al., 2021), natural extension to video domains (Bertasius et al., 2021), integration with other domains like language or speech (Radford et al., 2021), and easily combined with state-of-the-art visual self-supervised learning (He et al., 2021).

The core concept of patch-based models is to update patch-wise representations by alternating the computation *within* patches and *among* patches, called channel mixing and patch (or token) mixing, respectively. The design of patch mixing layers is widely investigated: the pioneering Vision Transformer (ViT; Dosovitskiy et al. (2021)) and its descendants (Touvron et al., 2021a;b) considered self-attention (Vaswani et al., 2017), while other works considered feed-forward (Tolstikhin et al., 2021), convolution (Trockman & Kolter, 2022), or pooling operation (Yu et al., 2021). However, most patch-based models use self-attention or feed-forward mixing layers, which minimizes the inductive biases from the model by employing all patches equally (Khan et al., 2021).

While this data-centric approach is effective on large-scale scenarios, recent works claim that incorporating inductive biases is still essential for patch-based models, especially when learned from limited data (Steiner et al., 2021). To this end, many approaches incorporated spatial inductive bias for patch-based models following the analogy of convolution: designing a patch mixing layer utilizing the location of patches (d'Ascoli et al., 2021; Dai et al., 2021; Wu et al., 2021a; Trockman & Kolter, 2022) or building an architecture that combines convolutional or pooling layers with patch mixing layers (Liu et al., 2021; Yuan et al., 2021b; Wang et al., 2021; Fan et al., 2021). However, both approaches focus on spatial inductive bias, which overlooks the sample-specific object structures. We provide additional discussion on related work in Appendix A.

---

[*]Equal contribution

**Contribution.** We propose ReMixer, a novel reweighting scheme for patch mixing layers leveraging the object structure of images. We demonstrate that ReMixer improves classification accuracy and background robustness of patch-based models, outperforming the models considering spatiality.

## 2 ReMixer: Object-aware Mixing Layer

The main idea of ReMixer is to strengthen the interaction of patches containing similar objects while regularizing the connection of different objects (and background). Intuitively, ReMixer improves the discriminability of objects (i.e., better classification) and reduces the spurious correlations between objects and backgrounds (i.e., robust to background and distribution shifts) by learning disentangled representations of objects. We first introduce a general framework of object-aware mixing layer in Section 2.1, then illustrate the specific instantiations for various architectures in Section 2.2.

### 2.1 Computing reweighting mask for ReMixer

The idea of patch-based models is to reshape a 2D image $\mathbf{x} \in \mathbb{R}^{H \times W \times C}$ into a sequence of flattened 2D patches $\mathbf{x}^0 \in \mathbb{R}^{N \times (P^2 C)}$, where $(H, W)$ is the resolution of original image, $C$ is the number of color channels, $(P, P)$ is the resolution of each image patch, and $N = HW/P^2$ is the number of patches. Patch-based models first convert the 2D patches into patch (or token) features $\mathbf{x}^1 := f_{\text{embed}}(\mathbf{x}^0) \in \mathbb{R}^{N \times D}$ with latent dimension $D$ using an embedding function $f_{\text{embed}}$, then update the patch features by alternating two operations: (a) patch mixing layers $f_{\text{mix}}^l : \mathbb{R}^N \to \mathbb{R}^N$ which mix the features among patches, and (b) channel mixing layers $g_{\text{mix}}^l : \mathbb{R}^D \to \mathbb{R}^D$ which mix the features among channels, where $l$ implies the operation of layer $l$. Formally, the $l$-th layer of patch-based model updates an input vector $\mathbf{x}^l \in \mathbb{R}^{N \times D}$ to an output vector $\mathbf{x}^{l+1} \in \mathbb{R}^{N \times D}$ following:

$$\mathbf{z}^{l+1} = [\mathbf{z}_{1:N,d}^{l+1}] = [f_{\text{mix}}(\mathbf{x}_{1,d}^l \; ; \; \mathbf{x}_{2,d}^l \; ; \; ... \; ; \; \mathbf{x}_{N,d}^l)] \tag{1}$$

$$\mathbf{x}^{l+1} = [\mathbf{x}_{n,1:D}^{l+1}] = [g_{\text{mix}}(\mathbf{z}_{n,1}^{l+1} \; ; \; \mathbf{z}_{n,2}^{l+1} \; ; \; ... \; ; \; \mathbf{z}_{n,D}^{l+1})] \tag{2}$$

where $\mathbf{z}_{n,d}$ and $\mathbf{x}_{n,d}$ denotes $n$-th patch, $d$-th channel value, and $\mathbf{z}_{1:N,d}^{l+1} \in \mathbb{R}^N$ and $\mathbf{x}_{n,1:D}^{l+1} \in \mathbb{R}^D$ denotes row-wise and column-wise subvector of $\mathbf{z}$ and $\mathbf{x}$, respectively.

We introduce ReMixer, a universal framework for improving patch mixing layers $f_{\text{mix}}$ by incorporating the object structure of images. To this end, we utilize the patch-wise labels $\mathbf{y} \in \mathbb{R}^{N \times K}$ where $K$ is the number of object classes. Using them, we compute the reweighting mask $\mathbf{M}^l \in \mathbb{R}^{N \times N}$ that strengthens the interaction of patches of similar objects while regularizing the connection of different objects and backgrounds. Formally, we set the $(i, j)$-th value of the reweighting mask $\mathbf{M}_{ij}^l$ as a reverse distance between the object labels of two patches $\mathbf{y}_i$ and $\mathbf{y}_j$:

$$\mathbf{M}_{ij}^l := \exp(-\kappa^{(l)} \cdot d(\mathbf{y}_i, \mathbf{y}_j)) \in (0, 1] \tag{3}$$

where $\kappa^{(l)} \geq 0$ is a learnable mask scale (scalar) parameter for each layer and $d(\cdot, \cdot) : \mathbb{R}^K \times \mathbb{R}^K \to \mathbb{R}$ is a distance function for object labels. We initialize $\kappa^{(l)}$ by zero for training, i.e., consider the full interaction initially then focus on the objects as $\kappa^{(l)}$ increases. We observe that the model sets higher $\kappa^{(l)}$ for lower layers and lower $\kappa^{(l)}$ for higher layers (especially, $\kappa^{(l)} = 0$ for the final layer) after training, i.e., ReMixer automatically attends the intra-object relations first then expand to the inter-object relations, which resembles the local-to-global structure of CNNs (see Table 3).

We aim to calibrate the $N \times N$ interaction of patch mixing layers using the reweighting mask $\mathbf{M}$. If the patch mixing layer $f_{\text{mix}} := \mathbf{L}_{\text{mix}}$ is linear, one can simply (element-wise) multiply the mask to get the masked linear mixer $\mathbf{M} \odot \mathbf{L}_{\text{mix}}$. While ReMixer can be applied to any patch layers in principle, one needs careful design to consider the nonlinear dynamics of each layer. We provide specific implementations of ReMixer on various representative models in the next section.

**Obtaining object labels.** We extract object labels from saliency (Voynov et al., 2021) or classification (Yun et al., 2021) models trained from other source datasets. See Appendix B for details.

### 2.2 ReMixer for ViT and Mixers

We briefly review three representative patch mixing layers: self-attention, feed-forward, and convolution, and describe the implementations of ReMixer for each layer.

**ReMixer for self-attention.** Self-attention (Vaswani et al., 2017) mixing layers update patch features by aggregating values with normalized importances (or attentions):

$$f_{\texttt{mix}}(\mathbf{x}) := \underbrace{\texttt{Softmax}(\frac{\mathbf{QK}^T}{\sqrt{D_K}})}_{\text{attention matrix } \mathbf{A}} \cdot \mathbf{V} \tag{4}$$

where $\mathbf{Q}$, $\mathbf{K}$, $\mathbf{V}$ are query, key, and value, respectively, which are linear projections of input $\mathbf{x} \in \mathbb{R}^{N \times D}$, given by $\mathbf{Q} := \mathbf{x} \cdot \mathbf{W}_Q \in \mathbb{R}^{N \times D_K}$, $\mathbf{K} := \mathbf{x} \cdot \mathbf{W}_K \in \mathbb{R}^{N \times D_K}$, and $\mathbf{V} := \mathbf{x} \cdot \mathbf{W}_V \in \mathbb{R}^{N \times D_V}$. Here, we compute $H$ independent attention heads and aggregate outputs for the final output of size $N \times D$ for $D = H \cdot D_V$. Recall that self-attention is basically a matrix multiplication of attention $\mathbf{A}$ and value $\mathbf{V}$ matrices, and one can (element-wise) multiply the reweighting mask $\mathbf{M}$ to the attention matrix to calibrate interaction. Then, we renormalize the masked attention $\mathbf{M} \odot \mathbf{A}$ to make the row-wise sum be 1 as the original self-attention. To sum up, ReMixer for self-attention is:

$$f_{\texttt{remix}}(\mathbf{x}) := [\tilde{\mathbf{A}}_{ij}] \cdot \mathbf{V} = [\frac{\mathbf{M}_{ij} \cdot \mathbf{A}_{ij}}{\sum_j \mathbf{M}_{ij} \cdot \mathbf{A}_{ij}}] \cdot \mathbf{V} \tag{5}$$

where $\tilde{\mathbf{A}}$ is the renormalized masked attention. We finally remark that patch-based models using self-attention often use the additional [CLS] token to aggregate the global feature. Here, we define the mask value between the [CLS] token and every other patch to be one and apply Eq. (5).

**ReMixer for feed-forward.** Feed-forward (or multi-layer perceptron; MLP) mixing layers update patch features with a channel-wise MLP. Since each channel is computed independently, we only consider a $N \times 1$ vector of a single channel. Then, the mixer layer is:

$$f_{\texttt{mix}}(\mathbf{x}) := \mathbf{W}_m \cdot \sigma(\mathbf{W}_{m-1} \cdot \sigma(\cdots \sigma(\mathbf{W}_1 \cdot \mathbf{x}))) \tag{6}$$

where $\mathbf{W}_1, ..., \mathbf{W}_m$ are weight matrices and $\sigma$ is a nonlinear activation. However, it is nontrivial to apply the reweighting mask $\mathbf{M}$ since $f_{\texttt{mix}}$ is nonlinear. To tackle this issue, we decompose the mixing layer $f_{\texttt{mix}}$ into a linear approximation $\mathbf{L}_{\texttt{mix}}^{\mathbf{x}} \cdot \mathbf{x} \approx f_{\texttt{mix}}(\mathbf{x})$ for a (possibly data-dependent) matrix $\mathbf{L}_{\texttt{mix}}^{\mathbf{x}} \in \mathbb{R}^{N \times N}$ and a residual term $f_{\texttt{mix}}(\mathbf{x}) - \mathbf{L}_{\texttt{mix}}^{\mathbf{x}} \cdot \mathbf{x}$. Here, we only calibrate the linear term but omit the residual term. Then, ReMixer for feed-forward is given by:

$$f_{\texttt{remix}}(\mathbf{x}) := \underbrace{(\mathbf{M} \odot \mathbf{L}_{\texttt{mix}}^{\mathbf{x}}) \cdot \mathbf{x}}_{\text{masked linear}} + \underbrace{(f_{\texttt{mix}}(\mathbf{x}) - \mathbf{L}_{\texttt{mix}}^{\mathbf{x}} \cdot \mathbf{x})}_{\text{residual}} \tag{7}$$

where $\odot$ is an element-wise product. While finding a good $\mathbf{L}_{\texttt{mix}}^{\mathbf{x}}$ is nontrivial in general, we found that a simple trick of dropping nonlinear activations gives an efficient yet effective solution:

$$\mathbf{L}_{\texttt{mix}} := \mathbf{W}_m \, \mathbf{W}_{m-1} \cdots \mathbf{W}_1 \in \mathbb{R}^{N \times N}. \tag{8}$$

We observe that this (somewhat crude) approximation performs well in practice. We also tried some data-dependent variants but did not see gain despite their computational burdens.

**ReMixer for convolution.** Convolutional mixing layers update patch features with a channel-wise 2D convolution. Similar to the feed-forward case, we only consider a single channel input $\mathbf{x}$ (which is $\mathbf{x}_{1:N,d}$ formally), reshaped as a $1 \times \bar{H} \times \bar{W}$ tensor where $(\bar{H}, \bar{W}) = (H/P, W/P)$ is the resolution of patch features. Then, the mixer layer is:

$$f_{\texttt{mix}}(\mathbf{x}) := \mathbf{W}_{\texttt{conv}} * \mathbf{x} \tag{9}$$

where $\mathbf{W}_{\texttt{conv}} \in \mathbb{R}^{1 \times 1 \times S \times S}$ is a kernel matrix with size $S$ and $*$ denotes convolution operation. Here, we consider the linearized version of kernel matrix (i.e., Toeplitz matrix) that substitutes the convolution to the matrix multiplication. Then, one can interpret the mixer layer as:

$$f_{\texttt{mix}}(\mathbf{x}) = \mathbf{W}_{\texttt{linear}} \cdot \tilde{\mathbf{x}} \tag{10}$$

where $\mathbf{W}_{\texttt{linear}} \in \mathbb{R}^{N \times N}$ is the corresponding matrix of $\mathbf{W}_{\texttt{conv}}$ and $\tilde{\mathbf{x}}$ is a reshaped tensor of $\mathbf{x}$ of size $N \times 1$, where $N = \bar{H} \cdot \bar{W}$. Here, one can directly multiply the reweighting mask $\mathbf{M}$ to define the ReMixer for convolution:

$$f_{\texttt{remix}}(\mathbf{x}) := (\mathbf{M} \odot \mathbf{W}_{\texttt{linear}}) \cdot \tilde{\mathbf{x}} \tag{11}$$

where $\odot$ is an element-wise product. We also compare ReMixer with the models using different kernel matrix for each channel, i.e., $\mathbf{W}_{\texttt{conv}} \in \mathbb{R}^{D \times 1 \times S \times S}$ (see Appendix D).

Table 1: ReMixer using various object labelers evaluated on the Background Challenge benchmark. '+' denotes the modules added to the baseline (not accumulated), and parenthesis denotes the gain of each module. ReMixer consistently improves the classification accuracy (↑) and background robustness (↓) of various patch-based models: DeiT, MLP-Mixer, and ConvMixer.

| | Patch labeler | Original (↑) | Only-BG-B (↓) | Only-FG (↑) | Mixed-Same (↑) | Mixed-Rand (↑) | BG-Gap (↓) |
|---|---|---|---|---|---|---|---|
| DeiT-S | - | 82.69 | 39.46 | 57.63 | 73.88 | 50.49 | 23.39 |
| + ReMixer (ours) | BigBiGAN | 83.88 (+1.19) | 36.59 (-2.87) | 60.10 (+2.47) | 75.95 (+2.07) | 54.07 (+3.58) | 21.88 (-1.51) |
| + ReMixer (ours) | ReLabel | 86.32 (+3.63) | 31.78 (-7.68) | 61.93 (+4.30) | 78.44 (+4.56) | 56.74 (+6.25) | 21.70 (-1.69) |
| MLP-Mixer-S/16 | - | 84.99 | 40.96 | 63.63 | 76.52 | 54.72 | 21.80 |
| + ReMixer (ours) | BigBiGAN | 86.30 (+1.31) | 36.64 (-4.32) | 66.15 (+2.52) | 78.47 (+1.95) | 58.54 (+3.82) | 19.93 (-1.87) |
| + ReMixer (ours) | ReLabel | 87.68 (+2.69) | 27.28 (-13.68) | 67.88 (+4.25) | 79.43 (+2.91) | 60.44 (+5.72) | 18.99 (-2.81) |
| ConvMixer-512/8 | - | 86.32 | 41.38 | 66.84 | 78.59 | 56.99 | 21.60 |
| + ReMixer (ours) | BigBiGAN | 86.35 (+0.03) | 37.09 (-4.29) | 70.03 (+3.19) | 80.27 (+1.68) | 59.19 (+2.20) | 21.09 (-0.51) |
| + ReMixer (ours) | ReLabel | 88.49 (+2.17) | 35.60 (-5.78) | 71.60 (+4.76) | 81.70 (+3.11) | 63.93 (+6.94) | 17.77 (-3.83) |

Table 2: Comparison of ReMixer (with ReLabel) vs. ConViT (spatial inductive bias). '+' denotes the modules added to the baseline (not accumulated), and parenthesis denotes the gain of each module.

| | Original (↑) | Only-BG-B (↓) | Only-FG (↑) | Mixed-Same (↑) | Mixed-Rand (↑) | BG-Gap (↓) |
|---|---|---|---|---|---|---|
| DeiT-S | 82.69 | 39.46 | 57.63 | 73.88 | 50.49 | 23.39 |
| + ConViT | 85.51 (+2.82) | 38.94 (-0.52) | 62.15 (+4.52) | 76.40 (+2.52) | 54.37 (+3.88) | 22.03 (-1.36) |
| + ReMixer (ours) | 86.32 (+3.63) | 31.78 (-7.68) | 61.93 (+4.30) | 78.44 (+4.56) | 56.74 (+6.25) | 21.70 (-1.69) |
| + ConViT + ReMixer (ours) | 88.20 (+5.51) | 30.79 (-8.67) | 66.82 (+9.19) | 79.16 (+5.28) | 60.52 (+10.03) | 18.64 (-4.75) |

Table 3: Learned mask scales $\kappa^{(l)}$ over layers.

| | Layer 1/4 | Layer 2/4 | Layer 3/4 | Layer 4/4 |
|---|---|---|---|---|
| DeiT-S | 1.571 | 0.783 | 0.945 | 0.287 |
| MLP-Mixer-S/16 | 0.744 | 0.000 | 0.001 | 0.061 |
| ConvMixer-512/8 | 0.871 | 2.347 | 1.498 | 0.001 |

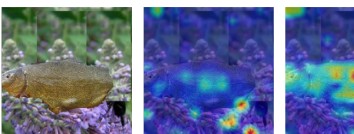

(a) Original  (b) DeiT-S  (c) ReMixer

Table 4: Saliency map visualization.

## 3 EXPERIMENTS

We first verify the efficacy of ReMixer in Section 3.2, outperforming both vanilla patch-based models and models considering spatial inductive bias. Then, we demonstrate the robustness of ReMixer on out-of-distribution datasets in Section 3.3. We also show that ReMixer outperforms the method that requires the same-level of supervision in Section 3.4, confirming that the remixing of patch mixing layers gives improvements. The experimental settings are in Section 3.1.

### 3.1 EXPERIMENTAL SETTINGS

**Models and training.** We apply ReMixer on three representative patch-based models: DeiT (Touvron et al., 2021a), MLP-Mixer (Tolstikhin et al., 2021), and ConvMixer (Trockman & Kolter, 2022), using self-attention, feed-forward, and convolutional patch mixing layers, respectively. Specifically, we use DeiT-S, MLP-Mixer-S/16, and ConvMixer-512/8. ConvMixer-512/8 has kernel size 9 and patch size 16 following MLP-Mixer-S/16 configurations, and we share the convolution kernel over channels by default, while we provide the unshared results in Appendix D. We follow the default training setup of (Touvron et al., 2021a), but use 256 batch sizes due to memory issues in our GPUs.

**Object labels.** We consider three object labels: first two are BigBiGAN (Voynov et al., 2021) and ReLabel (Yun et al., 2021), representative methods for binary saliency and multi-class prediction maps, respectively. We also test Bbox+GrabCut, which extracts binary masks from the ground-truth bounding boxes using the GrabCut (Rother et al., 2004) algorithm.

### 3.2 MAIN RESULTS

**Setup.** We train the models on the ImageNet-9 (Xiao et al., 2021a), which is a 9 superclass subset of ImageNet. We report the results on the Background Challenge (Xiao et al., 2021a) benchmark to evaluate the background robustness of models. Background Challenge contains 8 datasets: ORIGINAL (↑), ONLY-BG-B (↓), ONLY-BG-T (↓), NO-FG (↓), ONLY-FG (↑), MIXED-SAME (↑), MIXED-RAND (↑), and MIXED-NEXT (↑), where the upper or lower arrows indicate the model

Table 5: Test accuracy of ReMixer (with BigBiGAN) trained on ImageNet-9 and tested on out-of-distribution. '+' denotes the modules added to the baseline, and bold denotes the best results.

| | ImageNet-9 | ImageNetV2-9 | ReaL-9 | Rendition-9 | Stylized-9 | Sketch-9 |
|---|---|---|---|---|---|---|
| DeiT-S | 82.69 | 73.22 | 80.21 | 28.60 | 21.90 | 27.26 |
| + ReMixer (ours) | **83.88** (+1.19) | **75.68** (+2.46) | **82.37** (+2.16) | **29.43** (+0.83) | **24.64** (+2.74) | **27.93** (+0.67) |
| MLP-Mixer-S/16 | 84.99 | **76.57** | 82.70 | 34.25 | 25.54 | 37.07 |
| + ReMixer (ours) | **86.30** (+1.31) | 76.51 (-0.06) | **83.37** (+0.67) | **34.29** (+0.04) | **27.08** (+1.54) | **38.16** (+1.09) |
| ConvMixer-512/8 | 86.32 | 76.38 | 83.16 | 33.33 | 24.21 | 34.36 |
| + ReMixer (ours) | **86.35** (+0.03) | **76.97** (+0.59) | **83.56** (+0.40) | **33.81** (+0.48) | **24.94** (+0.73) | **35.72** (+1.36) |

Table 6: Test accuracy of ReMixer (with ReLabel) trained on ImageNet. We compare TokenLabeling (TL) and TL+ReMixer (Ours), verifying our remixing scheme on the fair comparison setting.

| | TL | TL + ReMixer |
|---|---|---|
| DeiT-S | 80.20 | 81.26 (+1.06) |
| DeiT-B | 81.17 | 82.18 (+1.01) |

should predict the class well or not, respectively. We also report BG-GAP ($\downarrow$) which measures the accuracy gap between MIXED-SAME and MIXED-RAND. We omit ONLY-BG-T, NO-FG, and MIXED-NEXT results for the brevity of presentation (see Appendix C for discussion).

**Results.** Table 1 shows that ReMixer consistently improves classification accuracy and background robustness over various patch-based models and object labels. Table 2 compares ReMixer with ConViT (d'Ascoli et al., 2021), a patch mixing layer with spatial prior. It verifies that the object-centric structure of ReMixer is more effective than spatiality, yet gives orthogonal benefits.

**Analysis.** Table 3 reports the mask scales $\kappa^{(l)}$ of trained models, averaged by each quarter of layers. The models set higher/lower $\kappa^{(l)}$ for lower/higher layers, i.e., see the objects first then expand its view, like the local-to-global structure of CNNs. ConvMixer sets low $\kappa^{(l)}$ for the early layers since it is hard to understand the objects due to the restricted view of convolution. Figure 4 visualizes the saliency maps (Chefer et al., 2021), verifying that ReMixer gives more object-centric view.

## 3.3 ROBUSTNESS OF REMIXER

**Setup.** We evaluate the robustness of ReMixer inferred on unseen out-of-distribution (OOD) data. To this end, we test the ReMixer trained on ImageNet-9 on various OOD datasets: 9 superclass (370 class) subset of ImageNetV2 (Recht et al., 2019), ImageNet-ReaL (Beyer et al., 2020), ImageNet-R (Hendrycks et al., 2021), ImageNet-Stylized (Geirhos et al., 2019), and ImageNet-Sketch (Wang et al., 2019), denoted by adding '-9' at the end.

**Results.** Table 5 shows the OOD generalization results. ReMixer performs well on OOD samples, confirming that both object annotators and learned masks are transferable. We use the BigBiGAN annotator since it gives more robust prediction results than ReLabel.

## 3.4 FAIR COMPARISONS ON LARGE-SCALE DATASET

**Setup.** We compare ReMixer with TokenLabeling (TL; Jiang et al. (2021)) on ImageNet. TL uses the same extra patch-level label with ReMixer. The experiment settings are same as Section 3.1.

**Results.** Table 6 presents the comparison of TL and TL+ReMixer (Ours). TL+ReMixer outperforms TL, demonstrating that the guided patch interaction of ReMixer drives the performance gain further.

## 4 CONCLUSION

We propose ReMixer, a novel object-centric framework to refine any existing patch-based models. We demonstrate the efficacy of ReMixer on ViT, MLP-Mixer, and ConvMixer, while showing superior (yet compatible) performance over prior works using spatial inductive bias. We hope ReMixer could inspire new research directions for patch-based models and object-centric learning.

ACKNOWLEDGMENTS

This research was supported by the Engineering Research Center Program through the National Research Foundation of Korea (NRF) funded by the Korean Government MSIT (NRF-2018R1A5A1059921). We thank Sihyun Yu, Jihoon Tack, Jaeho Lee, and Jongjin Park for constructive feedback.

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

# A  RELATED WORK

**Patch-based models.** Inspired by the success of Transformers (or self-attention; Vaswani et al. (2017)) in natural language processing (Devlin et al., 2019; Brown et al., 2020), numerous works have attempted to extend Transformers for computer vision (Khan et al., 2021). In particular, the seminal work named Vision Transformer (ViT; Dosovitskiy et al. (2021)) discovered that Transformer could achieve state-of-the-art performance, exceeding prior popular convolutional neural networks (CNNs; LeCun et al. (1998)). Thereafter, other studies revealed that different patch mixing layers such as feed-forward (Tolstikhin et al., 2021), convolution (Trockman & Kolter, 2022), or pooling (Yu et al., 2021) show comparable performance to self-attention, hypothesizing that the success of ViT comes from the patch-based architectures. Our work proposes an architecture-agnostic framework to improve the patch-based models by reweighting their patch mixing layers.

**Inductive bias for patch-based models.** Many patch-based models aim to remove inductive biases by using patch mixing layers without additional structures, e.g., self-attention. While they perform well on large data regimes, recent works reveal that inductive biases are still crucial for patch-based models, especially under limited data (Steiner et al., 2021). Consequently, extensive literature proposed approaches to incorporate additional structures for patch-based models, e.g., spatial structures of CNNs. One line of work aims to design patch mixing layers reflecting inductive biases. For example, ConViT (d'Ascoli et al., 2021) and CoAtNet (Dai et al., 2021) calibrate self-attention with spatial distance between patches, CvT (Wu et al., 2021a) and ConvMixer (Trockman & Kolter, 2022) utilize convolution operation for patch mixing, and AS-MLP (Lian et al., 2021) design a structured operation aggregating the values from different axises. Another line of work build an architecture that combines convolutional or pooling layers with patch mixing layers Liu et al. (2021); Yuan et al. (2021b); Wang et al. (2021); Fan et al. (2021); Heo et al. (2021); Yuan et al. (2021a); Xiao et al. (2021b). Our work falls into the first category; however, we leverage the object structure of images, unlike prior works focused on the spatial inductive bias. Using rich information, our proposed ReMixer outperforms ConViT and CoAtNet, where using both ConViT and ReMixer gives further improvements, implying that two methods contribute to the model differently (see Table 2). We also emphasize that ReMixer can be applied on any patch mixing layers under a common principle, unlike prior works designed for specific layers such as self-attention or feed-forward.

**Incorporating object structures.** Although objects are the atom of visual scenes, only a limited number of research has leveraged the object structure of images for visual recognition (e.g., classification). This is mainly due to two reasons: (a) the cost of collecting object labels and (b) non-triviality of reflecting object information to the black-box deep learning models. However, both challenges have been relaxed by the rapid advance of deep learning. First, the progress of supervised (He et al., 2017; Carion et al., 2020; Fang et al., 2021), weakly-supervised (Selvaraju et al., 2017; Chefer et al., 2021; Yun et al., 2021), and self-supervised (Voynov et al., 2021; Caron et al., 2021; Mo et al., 2021) detection significantly reduced the cost of object labels. We utilize the pretrained BigBiGAN (Voynov et al., 2021) and ReLabel (Yun et al., 2021) models for our experiments; one could also train weakly- or self-supervised object annotators on their datasets. Second, the patch-based models are well-suited with object information (unlike CNNs) as the patch embeddings preserve their spatial information (Raghu et al., 2021). Using this property, ReMixer adjusts the interaction of patch embeddings using object labels. ORViT (Herzig et al., 2021) also direct ViT to focus on the object regions by creating extra object tokens. However, their goal is to guide video Transformers to track the trajectory of objects and are less suited for image classification. On the other hand, TokenLabeling (Jiang et al., 2021) implicitly utilizes the object information by using them as additional supervision for patch embeddings; it provides an orthogonal gain from ReMixer (see Table 6). We finally note that several works (Locatello et al., 2020; Wu et al., 2021b; Kipf et al., 2021) aim to disentangle the object features explicitly. However, they do not scale yet to the complex real-world images due to the strong constraints in the model. In contrast, ReMixer can be applied to any existing patch-based models with minimal modification.

# B  OBTAINING OBJECT LABELS

One possible concern for ReMixer is the labeling cost of patch-wise object labels $\mathbf{y} \in \mathbb{R}^{N \times K}$. However, we claim that this cost is not a critical issue since one can utilize the pretrained machine annotators. Notably, the object labels extracted from the models trained on some (source) datasets are still helpful for different (target) downstream datasets (see Section 3.3). In the remaining section, we describe two types of machine annotators: binary saliency and multi-class prediction with discussion on their pros and cons.

**Binary saliency map.** We first consider binary saliency maps, i.e., indicating whether the given pixel is object or background. There is a tremendous amount of work on extracting saliency maps in a self-supervised (Voynov et al., 2021; Caron et al., 2021; Mo et al., 2021) or weakly-supervised (i.e., using class labels; Selvaraju et al. (2017); Chefer et al. (2021)) manner. We use the saliency model called BigBiGAN (Voynov et al., 2021), which finds the salient region using BigGAN (Brock et al., 2019) trained on the ImageNet (Deng et al., 2009) dataset. We average the pixel-wise saliency values in the patch to get a soft label $\mathbf{y}_n \in [0, 1]$, and use the $l_1$-distance (between the object labels of two patches) in Eq. (3).

**Multi-class prediction map.** We also consider multi-class prediction maps, i.e., pixel- or patch-level semantic segmentation. However, since segmentation labels are expensive, we utilize ReLabel (Yun et al., 2021), which predicts the dense label maps from an image classifier by applying the classifier on penultimate spatial features (i.e., before global average pooling). We use the NFNet-F6 (Brock et al., 2021) model trained on the ImageNet dataset to extract ReLabel map and apply region-of-interest (RoI) pooling (Girshick, 2015) to extract the patch label $\mathbf{y}_n \in \mathbb{R}^K$. Here, computing $l_p$-distance between the object labels of patches is expensive since it handles a $N^2 K$ tensor. Instead, we use the cosine distance, efficiently computed by matrix multiplication.

**Comparison of two approaches.** Multi-class prediction map contains richer semantics and thus provide more informative reweighing masks. As a result, ReLabel (used as the object labels for ReMixer) has shown better results than BigBiGAN in our experiments, especially when the downstream tasks are close to the source dataset, ImageNet. However, ReLabel is often prone to distribution shifts, while BigBiGAN provides consistent gain under the same setup. Intuitively, predicting the salient objects is easier to generalize than predicting classes. Thus, we suggest the users choose binary vs. multi-class labels following their robustness vs. in-distribution accuracy trade-off.

We finally remark that while we utilize the pretrained annotators (from different source datasets) for simplicity, one can also apply our method without external datasets. Recall that the annotators we consider are trained in an unsupervised or weakly-supervised manner; it does not require ground-truth dense supervision and can be solely extracted from the downstream dataset.

## C  DISCUSSION ON BACKGROUND CHALLENGE

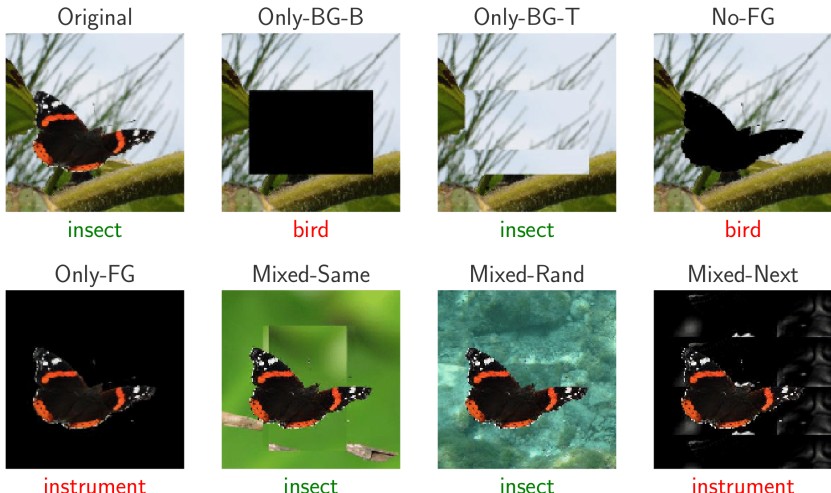

Figure 1: Datasets in the Background Challenge benchmark.

Figure 1 visualizes the datasets in the Background Challenge (Xiao et al., 2021a) benchmark, which provides various combinations of foregrounds and backgrounds: ORIGINAL (↑), ONLY-BG-B (↓), ONLY-BG-T (↓), NO-FG (↓), ONLY-FG (↑), MIXED-SAME (↑), MIXED-RAND (↑), MIXED-NEXT (↑). The upper or lower arrows indicate the model should predict the class well or not, respectively. We omit ONLY-BG-T, NO-FG, and MIXED-NEXT results for brevity of presentation and due to the following reasons:

- MIXED-NEXT shows almost the same trend with MIXED-RAND.
- NO-FG is controversial to be predicted or not since the image contains the black shape of the object.
- ONLY-BG-T accuracy is proportional to the ORIGINAL accuracy. However, we remark that ReMixer only increases the ONLY-BG-T accuracy a little while ORIGINAL a lot, compared to the baseline models.

## D  SHARING WEIGHTS FOR CONVMIXER

Table 7: Comparison of ConvMixer sharing (white line) and not sharing (gray line) kernels over channels. We add '-Full' to denote the latter model, which uses $D$ times for parameters for patch mixing layers where $D$ is the number of channels.

| | Original (↑) | Only-BG-B (↓) | Only-FG (↑) | Mixed-Same (↑) | Mixed-Rand (↑) | BG-Gap (↓) |
|---|---|---|---|---|---|---|
| ConvMixer-512/8 | 86.32 | 41.38 | 66.84 | 78.59 | 56.99 | 21.60 |
| + ReMixer (ours) | 88.49 (+2.17) | **35.60** (-5.78) | **71.60** (+4.76) | **81.70** (+3.11) | **63.93** (+6.94) | **17.77** (-3.83) |
| ConvMixer-512/8-Full | **88.67** (+2.35) | 40.20 (-1.18) | 70.72 (+3.88) | 81.31 (+2.72) | 62.84 (+5.85) | 18.47 (-3.13) |

We adopt ConvMixer (Trockman & Kolter, 2022) to share the convolution kernels over channels to reduce memory and computation. Remarkably, ReMixer applied on ConvMixer is comparable or even outperforms ConvMixer-Full, which uses different kernels for each channel, i.e., uses $D = 512$ times parameters for patch mixing layers.

