# OpenReview forum: "ReMixer: Object-aware Mixing Layer for Vision Transformers and Mixers"
_ICLR.cc/2022/Workshop/OSC — ICLR2022 OSC  Poster_

### Official Review · Reviewer_SVkh · 2022-03-04
**Good paper with clear motivation and extensive experiments**

**Rating:** 2
**Confidence:** 3

**Review:**

Pros:

1. Clear motivation. There are many work recently to improve patch-based models. Object-centric is one of the approaches. The writing and organization of the submission is good.
2. Through extensive experiments, ReMixer shows its potential on a variety of model architectures and downstream tasks. In most cases, the improvement is nontrivial.
3. Ablation study on obtaining object labels, sharing weights, etc is complete, which makes the conclusion convincing.

Some comments:

1. When it comes to hierarchical transformers like Swin-transformer, how to apply ReMixer?
2. Is it necessary to do ReMixer for every transformer block? Maybe just the first several blocks or even just in tokenization stage may be enough. Is there any analysis regarding the position of using ReMixer?
3. Object-centric learning usually leads to better robustness, it also brings improvement for localization tasks. I'm curious to know how ReMixer performs for downstream object detection and segmentation tasks.
4. Is there any failure cases? For example, after applying ReMixer, which class improves most and which class might drop accuracy?

Overall, this is a nice paper with good experimental results. I'm inclined to accept.

---

### Official Review · Reviewer_VHJv · 2022-03-12
**Review for ReMixer**

**Rating:** 2
**Confidence:** 2

**Review:**

This paper decompose the representations of the operators into patch mixing and channel mixing, and propose an interesting idea of using reweighting mask to improve the representations of the cross-patch mixing. The proposed Remixer leverages the differences between patches. It can be applied to self-attention, linear and conv layers. The experiments demonstrate the effectiveness of the proposed approach on background challenge.

The proposed  method has very limited domain to apply due to requirement of patch wise labels. Therefore, it can't be generalized to other major CV problems. The experiments are not comprehensive. It is unclear whether the improvement is due to the extra supervision or increasing the compute or mainly be the proposed representations.

I am around the borderline. Despite the weakness of this work, there is no significant issue for this work (e.g. technically incorrect or off-topic). I rate this paper as "Accept".

---

### Decision · Program_Chairs · 2022-03-21

**Decision:**

Accept (Poster)

**Comment:**

The reviewers agree the paper should be accepted at the workshop. Congratulations!